# Systematic Review of Irreversible Electroporation Role in Management of Locally Advanced Pancreatic Cancer

**DOI:** 10.3390/cancers11111718

**Published:** 2019-11-03

**Authors:** Stefano Lafranceschina, Oronzo Brunetti, Antonella Delvecchio, Maria Conticchio, Michele Ammendola, Giuseppe Currò, Tullio Piardi, Nicola de’Angelis, Nicola Silvestris, Riccardo Memeo

**Affiliations:** 1Department of Emergency and Organ Transplantation, University “Aldo Moro” of Bary, 70124 Bary, Italy; antodel88@libero.it (A.D.); maria_cont@hotmail.it (M.C.); drmemeo@yahoo.it (R.M.); 2Medical Oncology Unit, IRCCS Istituto Tumori “Giovanni Paolo II 2”, 70124 Bari, Italy; dr.oronzo.brunetti@tiscali.it (O.B.); n.silvestris@oncologico.bari.it (N.S.); 3Department of Health Science, General Surgery, Magna Graecia University, Medicine School of Germaneto, 88100 Catanzaro, Italy; michele.ammendola@libero.it (M.A.); currog@unime.it (G.C.); 4Department of Surgery, Hôpital Robert Debré, University of Champagne-Ardenne, 51100 Reims, France; tullio.piardi@gmail.com; 5Department of Digestive and Hepato-Pancreato-Biliary Surgery, Henri Mondor University Hospital, AP-HP, Université Paris-Est Créteil (UPEC), 51 Avenue du Maréchal de Lattre de Tassigny, 94010 Créteil, France; nic.deangelis@yahoo.it; 6Department of Biomedical Sciences and Human Oncology, University of Bari ‘Aldo Moro’, 70124 Bari, Italy

**Keywords:** pancreas, locally advanced, pancreatic cancer, irreversible electroporation

## Abstract

Background: Ablative techniques provide in patients with locally advanced pancreatic cancer (LAPC) symptomatic relief, survival benefit and potential downsizing. Irreversible Electroporation (IRE) represents potentially an ideal solution as no thermal tissue damage occurs. The purpose of this review is to present an overview on safety, feasibility, oncological results, survival and quality of life improvement obtained by IRE. Methods: A systematic search was performed in PubMed, regarding the use of IRE on PC in humans for studies published in English up to March 2019. Results: 15 original studies embodying 691 patients with unresectable LAPC who underwent IRE were included. As emerged, IRE works better on tumour sizes between 3–4 cm. Oncological results are promising: median OS from diagnosis or treatment up to 27 months. Two groups investigated borderline resectable tumours treated with IRE before resection with margin attenuation, whereas IRE has proved to be effective in pain control. Conclusions: Electroporation is bringing new hopes in LAPC management. The first aim of IRE is to offer a palliative treatment. Further efforts are needed for patient selection, as well as the use of IRE for ‘margin accentuation’ during surgical resection. Even if promising, IRE needs to be validated in large, randomized, prospective series.

## 1. Introduction

Pancreatic cancer (PC) remains a highly lethal disease. Currently, it is the fifth leading cause of death from cancer in men and the fourth in women in Italy. It has a dismal prognosis, with long-term survival rates of 5–6% at 5 years [1]. The only available potential cure for PC remains surgical resection with microscopically negative margins (R0) that offers the best chance for long term survival, but only the 15–20% of patients presenting PC are effectively candidates for resection [2]. This happens because at the time of presentation, due to a diagnostic delay, about the 30% of patients present locally advanced unresectable tumours, and 50% present metastatic disease; in all about the 80% of patients are not candidates for surgical resection [3]. For those patients that go onto resection, the 5-year survival ranges from 15–20%, whereas the 5-year survival for all PC patients combined is only 3% [2]. Multiple and varied factors lead to the overall dismal prognosis of PC, making its management challenging. Those factors include nonspecific symptoms that lead to delayed diagnosis, biological aggressiveness, which is resistant to chemotherapy, and surgical consideration that can be technically demanding. In this context there is a subset of patients with locally advanced pancreatic cancer (LAPC) which is less straightforward. LAPC evolves without evidence of distant macro-metastasis, and on macroscopic level is represented by two subclasses: borderline resectable and unresectable, depending on surrounding vascular involvement (Superior mesenteric artery and vein, celiac axis, hepatic artery, portal vein). The extent of vascular involvement and the possibility of their reconstruction define whether the LAPC is deemed borderline resectable or unresectable. For this reason, the definition of resecability has historically been vague and submitted to subjective interpretation (imaging, technical/surgical ability and overall institutional experience). Currently there are three definition of borderline resectable pancreatic cancer (BRPC) and LAPC within international guidelines: American Hepato-Pancreatico-Biliary Association (AHPBA), National Comprehensive Cancer Network (NCCN) and MD Anderson Cancer Center. The most commonly applied and cited classification is the one of NCCN: Borderline Resectable Tumours that have a contact superior to 180° with the portal vein (PV) or the superior mesenteric vein (SMV) or lower than 180° in the presence of thrombi or irregularity of the vessel wall; lack of interest in the celiac trunk; less than 180° involvement of the superior mesenteric artery (SMA) or any involvement of the common hepatic artery (CHA). LAPC are instead defined by occlusion or by the impossibility of reconstruction of PV or SMV; encasement or contact with the aorta; encasement of the SMA. To further confirm the validity of NCCN guidelines regarding BRPC and LAPC, the international study group for pancreatic surgery (ISGPS) adopted the same definition and guidelines.

The main goal of neoadjuvant settings and ablation therapies is to increase the amount of patients eligible for curative-intent surgery through a downsizing and less vessels involvement. Previously, the management of the LAPC patients foresees the use of gemcitabine based on chemotherapy in association or not with radiotherapy, achieving marginal benefits in terms of overall survival [4]. More recently, the use of new chemotherapeutic associations such as gemcitabine/nab paclitaxel and FOLFIRINOX (5-fluorouracil, leucovorin, irinotecan and oxaliplatin) as neoadjuvant setting for LAPC have increased the number of patients who can benefit from curative surgery among all LAPC patients. Also patients treated with FOLFIRINOX have a median overall survival of 24 months respect to those patients treated with gemcitabine (6–13 months) in recent studies [3]. Despite this, a large number of patients with locally advanced tumours remains ineligible for curative intent of surgical resection. It is precisely for these patients that the possibility of applying ablative techniques as an alternative has made its way over the years. Radiotherapy plays a role in the localized control of LAPC, with the aim to improve the local control and achieve in some cases tumour downstaging.

Currently patients with unresectable LAPC indeed have a poor median overall survival of 6–11.5 months in the majority of prospective clinical trials despite advances in chemotherapy, radiation therapy and chemoradiation therapy [5,6]. In these patients, after the induction therapy, ablative techniques such as radiofrequency ablation (RFA), microwave ablation (WMA), high intensity focused ultrasound (HIFU), cryoablation and irreversible electroporation (IRE) can provide symptomatic relief, survival benefit and potential downsizing. Nevertheless many of those procedures induce thermal injury to the pancreatic and bile duct that can result in fistulae or bile leaks, respectively, and thermal injury to adjacent vessels can result in significant bleeding (Table 1) [7].

IRE is a relatively new procedure which represents a potentially ideal solution for the ablative treatment of LAPC as no thermal tissue damage occurs, thus avoiding vessels or duct injury. Behind the possible success of this recently introduced technique in clinical practice, there is a study of the effect of electrical stimulation on cell membrane. When electric pulses are applied to cells, two different phenomena can occur: reversible electroporation and irreversible electroporation, both used in clinical practice. Reversible electroporation will increase cell membrane permeability and open an access route for molecules that are too big to cross the cell membrane such as DNA or RNA, or facilitates cell enter by hydrophilic molecules useful for applying of electrochemoporation (such as bleomycin or cisplatin). Those molecules once crossed the cell membrane, develop their effect in the resealing and intact cells (due to the increased permeability of the cell membrane, chemotherapeutic agents can pass into the cells and induce the mitosis death of cells in the targeted tissue). In contrast, irreversible electroporation is a non-thermal tissue ablation technology that by very short pulses of an high voltage current (maximum 3000 volts delivered in 70–80 microseconds), low energy create multiples microscopic holes within the cell membrane in order to make it permeable irreversibly. Cell membrane disruption leads to cellular apoptosis of pancreatic pathologic tissue within the ablation area, due to interference with homeostatic mechanism, preserving the underlying matrix, vessels and biliary ducts possibly included in the ablation area. This technique is therefore completely different from thermal ablation techniques. When IRE is performed, temperatures remain less than 50 °C, so IRE does not suffer from the heat-sink effect, collateral damage on surrounding tissues and does not cause coagulation necrosis [5,10]. Nevertheless, IRE treated lesions often show a centre of white coagulation surrounding the electrodes, histologically characterized by streamlined cytoplasm and pyknotic nuclei as happens for thermal necrosis caused by an increase in temperature during treatment [11]. The purpose of this review is to present an overview on safety, feasibility, oncological results, survival and quality of life improvement obtained by IRE as treatment of unresectable LAPC and borderline resectable LAPC.

## 2. Materials and Methods

A systematic search was performed in PubMed, regarding the use of IRE on PC in humans, using the search words ‘electroporation AND pancreas OR electroporation AND pancreatic OR irreversible electroporation OR IRE’ for studies published in English up to March 2019. All the titles and abstracts of those studies identified in the initial search were screened to identify those reporting on patients with unresectable or borderline resectable tumour undergoing ablation. We identified additional studies through hand searching of bibliographies from primary studies, key journals and review articles. We included also case reports. Endpoints of the search were to investigate safety, oncological results, complications, survival and quality of life in terms of symptoms control by using that technique.

Variables extracted from each study, where available, included: number of patients, demographic data (age and sex), tumour histology, size of the lesion, extent of disease (borderline or locally advanced or metastatic), operative approach (open, percutaneous, laparoscopic), associated therapies, complications, mean follow-up and survival (overall survival was defined as the interval from diagnosis or where available from IRE administration) and treatment response. Morbidity and mortality were also evaluated (morbidity was defined as the occurrence of any type of adverse event after IRE). After reviewing the studies that met the inclusion criteria, data were extracted from each individual study as proposed by the principles of the systematic review.

The patients performed diagnostic investigation including ultrasound (US), Computed Tomography (CT) and/or Magnetic Resonance Imaging (MRI), which revealed pancreatic lesions suspicious for PC confirmed by histological examination in almost all the studies. Locally advanced tumour was the general inclusion criterion, although Kluger et al. [7] treated three patients who had neuroendocrine tumours. All the patients examined underwent chemotherapy or chemoradiotherapy before undergoing IRE.

## 3. Results

For the identification of eligible studies, we used a flow diagram that schematically represents the article selection process (Figure 1). Altogether 15 original studies embodying 691 patients with unresectable LAPC who underwent IRE were included (Table 2). Whereas in Table 3 all the current clinical trials on Irreversible Electroporation for LAPC actually ongoing. Regarding the studies: eight reports were retrospective single center studies [12,13,14,15,16,17,18,19] and the remaining seven studies were prospective, single or multicenter studies [20,21,22,23,24,25,26] for a total of 15 selected studies. In those selected studies, the irreversible electroporation was applied in three different ways: a total of 392 patients underwent open (56.7%), percutaneous (275 patients, 39.8%) or laparoscopic (24 patients, 3.5%) procedures. 

Twelve studies reported data on the age of patients: the average age of patients among all those studies was 63.7 years. Approximately 80% of LAPC were located in the pancreatic head, neck or uncinated process. Only 10% of LAPC were located in the body or tail of pancreas. Tumour size ranged from 2.8 to 4.5 cm and median tumour size was 3.58 cm [12,13,14,15,16,17,19,23,24,25,26]. Cumulative overall morbidity was 30.5% (204 patients, according to the Clavien-Dindo classification).

Ten studies reported data on the application of preoperative chemotherapy: with the exception of the studies by Yan et al. [17] and Lambert et al. [15], approximately 70% of patients in each study received preoperative chemotherapy [14,16,17,18,20,23,24,25,26]. 

Nine studies reported data on the application of preoperative radiotherapy: in five studies less than 20% of patients received preoperative radiotherapy [17,18,24,25,26] while in other four studies the percentage does not exceed the 50% [14,16,20,23].

Cumulative mortality rate after IRE was 3.4% (19 patients), some studies reported no deaths in the post-procedural period. During follow up, median OS after IRE varied from 10 to 27 months.

## 4. Discussion

Irreversible electroporation therefore seems to have advantages over other ablative techniques, and can be preferred on those lesions near vessels or bile ducts. From the histological point of view, after IRE early changes occur in the target tissue after already 30 min, while the macroscopic changes are slower: those occur with more delay and become noticeable only in the weeks following irreversible electroporation [27,28,29]. From a practical point of view, the application of this procedure requires to be carried out under general anaesthesia with complete neuromuscular block to thereby reduce muscle contractions caused by the electrical pulses of the stimulation. Also this technique makes use of the possibility of surrounding the neoplastic mass with a number of needles ranging from two to six. The choice of the number of needles used during the procedure depends on size and shape of the target lesion: the formation pattern for the IRE needles must be based on tumour-specific properties with consideration of surrounding structures. Also important is the distance to be maintained during the use between the needles, which must not exceed 2.5 cm and must not be less than a centimetre for the technique to be effective. All needles must be also parallel to each other. That last technical need highlights a possible difficulty in positioning the electrodes correctly, which requires skills and experience often inherited from the use of other ablative methods, also with the use of multiple needles. It is also necessary to acquire skills in the field of ultrasound because a very precise ultrasound guide is required. This is whether it is a percutaneous procedure or an intraoperative procedure (laparoscopic or open). In the case of percutaneous procedure, the CT support can also be used to guide the positioning of the needles in order to avoid puncturing the surrounding organs accidentally. However, given the thinness of the needles used (22 Gauge), it is conceivable in some cases that are particularly difficult to positioning, to trans gastric or trans hepatic approaches. Regarding the proximity at important vessels, a minimum safety distance of 2 mm is recommended to avoid the risk of damage by burns. IRE can therefore be used with different approaches. If carried out by surgical teams, the open approach is usually preferred, which has the advantage of firstly verifying the presence or absence of peritoneal carcinosis and therefore of being able to positioning the needles parallel to the mesenteric vessels if they were involved in the LAPC (parallel positioning to the vessels when involved, its effectiveness has been improved). On the other hand the percutaneous approach has two important advantages: firstly the less invasivity in respect to the surgical approach and secondly, the possibility of positioning the needles even under CT guidance especially those that are near the mesenteric vessels. IRE is not free from contraindications, in particular it cannot be applied in case of cardiac arrhythmias as it could interact with myocardial contraction mechanisms, previous heart failure and active coronary disease. In addition to those cardiovascular contraindications, there is still epilepsy although IRE has not been shown to cause brain stimulation. From a clinical point of view, patients complain of abdominal pain from one up to three days after the procedure due to the development of the procedure of a mild acute pancreatitis with possible finding of very few laboratory variations. More severe complications may also occur such as more severe acute pancreatitis, portal or mesenteric thrombosis, the development of a pancreatic fistula, a bile leak, perforations of the gastro-enteric tract especially at the duodenal or transverse colon level, haemorrhages especially from the superior mesenteric artery. The actual risk of death is mainly linked to duodenal perforations or severe portal thrombosis [29]. 

Portal vein thrombosis is a rare but serious postoperative complication of irreversible electroporation. Although the mechanism is still unclear, literature studies about occurrence of the portal vein thrombosis report that it is associated with three factors: endothelial cell injury, slow blood flow and hypercoagulable state of the blood. In pancreatitis as can happen after IRE, various inflammatory mediators are released with concomitant thrombosis, resulting in high levels of IL-6, IL-8 and TNF. These factors stimulate the hepatic cells, which in turn produce large quantities of C-reactive protein (CRP). CRP is a sensitive indicator of the severity of inflammation. 

Clinical signs of portal vein thrombosis are often subtle and similar to those observed in postoperative pancreatitis. Early diagnosis of portal vein thrombosis therefore become essential: It can be achieved by using CT, MRI or Doppler US, and it might provide clinicians an opportunity for intervention before severe damage occurs. 

To relieve inflammatory response and post-procedural mild/acute pancreatitis, once the severity of pancreatitis is defined, it is important to intervene on several fronts. Supportive care including resuscitation with isotonic intravenous fluids, pain control and mobilization should be the mainstay of treatment. Nutrition care with early oral/enteral feeding in patients with acute pancreatitis should be considered since it is no longer associated with adverse effects and maybe associated with substantial decreases in pain, opioid usage and food intolerance while prophylactic antibiotics are not recommended in patients with mild or severe acute pancreatitis [30].

Based on those principles, numerous studies have been focused on its safety and feasibility. Despite of those assumptions, the average morbidity rate in the studies examined is 30%, and it can reach up to 59% in laparoscopic arm of Martin et al. study [22]. The list of possible adverse events include a miscellaneous number of complications, such as pancreatic fistula, venous and arterial thrombosis, pseudoaneurysm, pancreatic abscess (the majority of the complications, are consequence of an uncontrolled heating of the structures surrounding the tumour, rather than a direct lesion caused by the tip of the probe used). The average mortality rate of those studies is instead of the 3%. These numbers highlight the importance of the learning curve and the need to adopt IRE only in specialized centres. We are still far from saying that IRE is a standardized technique that can be used on a large scale.

Among all possible applications, as reported by the studies in Table 2, the open technique is the most adopted approach probably because it is safer than percutaneous or laparoscopic approach, and it allows more accurate probe placement Martin et al. [22] used the laparoscopic approach with an average increase in the percentage of complications (59.2%) without benefits in terms of overall survival (20.2 months from diagnosis). 

In order to best valorise the features of this technique it is necessary to plan its use with a pre-operative study as accurate as possible. As emerged from those studies, there is a theoretical dimensional cut-off for IRE: it works better on tumour sizes between 3–4 cm [14,15,19,23,25]. The extent of the ablation zone and its effectiveness are influenced by various parameters, including the diameter of the electrodes, the inter-electrodes distance and strength of the electric field. Naranyan et al. reported that tumour size was the only factor associated with OS in univariate [HR 0.43, 95% confidence interval (CI) 0.20–0.94; *p* = 0.035] and multivariate analysis: patients with tumours less than or equal to three centimetres had an advantage in terms of survival (*p* = 0.017). It is also important to take into account the difficulties of CT and MRI in measuring the response to the treatment: particularly in patients who have undergone radiation therapy before subjecting to IRE; and also the limitation to visualize potentially small (1–2 mm) tumour deposits incompletely [2]. Computed tomography to assess initial efficacy should be performed no earlier than three months after surgery because of the oedema and ongoing apoptosis seen in the early postoperative period.

The analysis of oncological results of IRE seems to be promising, with median OS from diagnosis or treatment of 27 months [16,18]. On the other hand, there are studies that report worse prognosis (15–17 months) [13,15,24,25,26]. It is possible that the variability of those data depends on the different designs of the studies, the expertise of single centres and the selection criteria of patients. The leading goals of many studies examined is safety and feasibility even before oncological outcomes. Interestingly Martin et al. [23] and Kluger et al. [13] also investigated a subgroup of patients with borderline resectable tumours that underwent resection with margin attenuation with IRE. The standard of care in this setting is still upfront surgery, as it is the only option for possible cure. Nevertheless, neoadjuvant chemotherapy in patients with borderline resectable pancreatic cancer is sometimes used, as it may increase the likelihood of achieving negative resection margins (R0), treat micrometastatic disease and it may also decrease the need for vascular reconstruction.

In several studies, LAPC patients underwent neoadjuvant therapy before IRE [12,13,14,19,20,23,24,25,31,32]. Leen et al. [17] considered several regimens (FOLFIRINOX, gemcitabine plus capecitabine, gemcitabine plus platinum, and gemcitabine alone) enrolling 28, 25, 12, and 10 patients, respectively, in the absence of data concerning survival. IMPALA study reported interesting results obtained with the use of neoadjuvant FOLFIRINOX before IRE with mOS and 1 year OS of 16 months and 78%, respectively [25]. In this study, patients with poor performance status received gemcitabine alone even if no data are reported for these patients in terms of survival. In Martin’s study [23], 200 patients received FOLFIRINOX or gemcitabine-based chemotherapy before IRE or resection plus IRE on positive margins. The authors pointed out the possibility of using FOLFIRINOX regimen or modified FOLFIRINOX (a more manageable regimen), since these two treatments achieved better oncological and surgical outcomes. Nevertheless, the authors did not compare the survival of patients treated with those regimens with the ones treated with a gemcitabine-based regimen. So far, there are no data on primary treatment which could improve IRE outcomes in this study. In the study carried out by He et al. [31] FOLFIRINOX and gemcitabine based chemotherapy were used before randomization to IRE or radiotherapy. No statistical significant differences in terms of survival had been achieved in the comparison between the primary treatments. On the contrary, 2-year OS rates were 53.5% and 20.7% (*p*  =  0.011) with 2-year PFS rates of 28.4% and 5.6% (*p*  =  0.004) for patients after treated with IRE and radiotherapy, respectively. Also, in a study conducted by Huang et al. [32] patients underwent neoadjuvant chemotherapy followed by IRE. It has been interesting to see that patients treated with TS-1 based chemotherapy showed a better survival compared to patients treated with gemcitabine (28.7 vs. 19.1 months, respectively, *p* = 0.04). Those results, however, should take into account that the study was conducted on an Asiatic population. In conclusion, a primary treatment could improve the clinical and radiological response of LAPC PDAC to IRE. Since FOLFIRINOX is one of the most active treatments in those patients, many Authors chose this regimen. 

Moreover, this setting can identify those patients whose disease will progress to metastatic disease, and thus they will be spared from unnecessary surgery. Those results underline the necessity of focusing studies on the possibility of using this technique during pancreatic resection for borderline LAPC as an accessory treatment for the pancreatic shear. Equally important to emphasize five cases of down staging with R0 resections of LAPC previously treated with percutaneous IRE described in two studies [6,24]. Even if it is not statistically significant, this is encouraging, especially if it is compared with LAPC population of all considered studies even because resection margin status is independently associated with long-term survival.

The data available concerning the role of adjuvant systemic chemotherapy after IRE is still controversial. In a preclinical study IRE enhanced gemcitabine delivery in primitive lesions [33]. In particular, gemcitabine concentration in an in vivo model of pancreatic tissue was higher in mice receiving IRE compared to those receiving gemcitabine alone (13,567 ng/mL vs. 4126 ng/mL; *p*  =  0.0009). Furthermore, in mice receiving IRE, pancreatic gemcitabine levels were higher than liver and serum levels. These data support the evaluation of adjuvant gemcitabine-based treatments after IRE. In Mansson’s study [14] patients treated with adjuvant therapy after IRE did not achieve a better survival compared with those who did not receive it. On the contrary, Belfiore et al. [6] underlined that their patients received gemcitabine-based chemotherapy. In any case it should be considered that all these studies are not randomized trials. So far, the authors treated their patients for their clinical practice merging therapies according to the expert opinions of institutional multidisciplinary boards. Furthermore, since FOLFIRINOX achieved significant results in terms of survival in adjuvant treatment compared to gemcitabine [34], this treatment has been proposed as adjuvant treatment after IRE, as was suggested in the IMPALA study [23]. Regarding the analysis of quality of life, IRE has instead proved to be an effective pain control tool with a significant reduction in the use of narcotic [19].

## 5. Conclusions

Electroporation is bringing new hopes in LAPC management. The aim of IRE is firstly to offer a palliative treatment, possibly with a mini-invasive approach, to LAPC patients with a good performance status who have already been treated by a consolidative systemic therapy. Most of studies are based on non-randomized series but have shown that IRE is promising in terms of overall survival (even if overall survival varied widely between different studies). Considering the possible complications, even serious, of whom they might be cause of, is important to emphasize that the decision supporting the use of IRE in LAPC should be made through experienced and highly specialized multidisciplinary teams. Further efforts are also needed to address patient selection, as well as the use of IRE for ‘margin accentuation’ during surgical resection [26]. It is also necessary to consider that it is an expensive technique with risks of complications and an associated mortality rate of 3%. Even if promising, IRE needs to be validated in large, randomized, prospective series. Moreover, it should be mandatory to identify the best neoadjuvant regimen to use before IRE with an improvement of oncological outcomes. Anyway, the future knowledge of pancreatic carcinogenesis [35] from its initiation within a normal cell until the time of metastasis could bring to new effective therapies for this malignancy, enhancing also this setting.

## Figures and Tables

**Figure 1 cancers-11-01718-f001:**
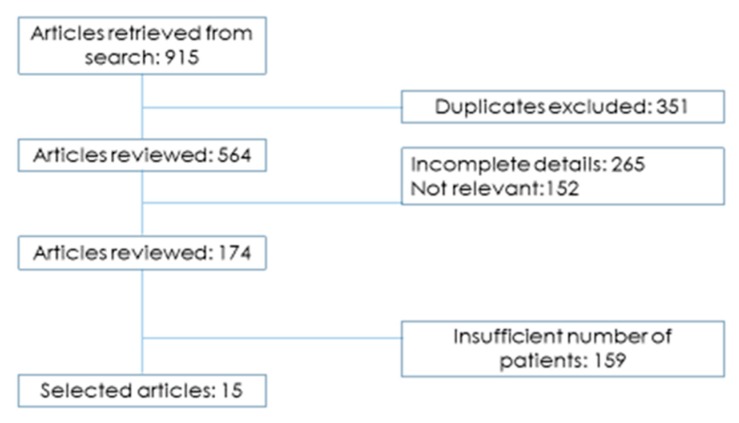
Study search strategy.

**Table 1 cancers-11-01718-t001:** Comparative analysis of literature data about ablative techniques and QoL in LAPC treatment [5,8,9].

Ablative Techniques	Target Patients	Ablation Technique	QoL Improvement	Complications
***HIFU***	LAPC Metastatic	Extracorporeal	Pain relief, Opioid intake, Better survival	Pancreatic fistula, gastric ulcers, pseudocyst formation, hematologic disorders
***RFA***	LAPC	Percutaneous Intraoperative	Pain relief	Heat damage, Hemorragies, acute pancreatitis, pancreatic fistula, biliary or duodenal burn, vein thrombosis
***IRE***	LAPC	Percutaneous Intraoperative	Pain relief, Opioid intake, Better survival	Pancreatic fistula, mild pancreatitits, vessel thrombosis, pseudoaneurysm
***Cryoablation***	LAPC Metastatic	Percutaneous Intraoperative	Mild pain relief	Severe pancreatitis, bleeding, pancreatic fistula, delayed gastric emptying
***Microwave***	LAPC	Percutaneous Intraoperative	Few reports	Mild pancreatitis, pancreatic fistula, minor bleeding

**Table 2 cancers-11-01718-t002:** Representative studies (at least 10 cases) reporting results of IRE on LAPC. (d): time from diagnosis, (t): time from treatment.

References	Pts (*n*)	Age	T Size (cm)	Prior RT (*n*)	Prior Cht (*n*)	Technique	Complication Rate (%)	Median OS (mo)	Mortality (%)
Belfiore [12]	29	68.5	4.5	NA	NA	Percut	0	14 (d)	NA
Kluger [13]	50	68.2	3.2			Open	24.5	12 (t)	6
Dunkl-Jacobs [20]	65	NA	NA	37	43	Percut/Open	20	NA	NA
Mansson [14]	24	65	3.5	10	22	Percut	45.8	17.9 (d)	4
Lambert [15]	21	68.2	3.9	NA	5	Percut/Open	23.8	10 (t)	0
Martin [21]	27	61	3.0			Percut/Open	33	17.9 (d)	4
Martin [22]	54	61	3.2			Open/VLS	59.2	20.2 (d)	2
Martin [23]	200	62	2.8	77	130	Open	37	24.9 (d)	2
Paiella [24]	10	66	3	4	10	Open	10	15.3 (d)	0
Vogel [25]	15	NA	NA	0	9	Open	53	16 (d)	13
Narayanan [16]	50	62.5	3.2	30	50	Percut	20	27 (d)	6
Yan [17]	25	58	4.2	3	1	Open	36	NA	4
Leen [18]	75	63.4	NA	4	75	Percut	25	27 (t)	0
Zhang [19]	21	NA	3.5	NA	NA	Percut	nr	NA	NA
Scheffer [26]	25	61	4	0	13	Percut	40	17 (d)	0

**Table 3 cancers-11-01718-t003:** Current Clinical trials on Irreversible Electroporation for LAPC.

Trial ID	Title	Phase	Number of Patients	Countries
**NCT02791503**	CROSSFIRE Trial: Cross atlantic randomized controlled trial comparing outcome in survival after systemic plus focal therapy for inoperable pancreatic carcinoma: RT vs. IRE	2–3	138	The Netherlands
**NCT03080974**	Immunotherapy and Irreversible Electroporation in the treatment of advanced pancreatic adenocarcinoma	2	10	USA
**NCT02718859**	Study of the combined therapy of IRE and Natural Killer Cells for advanced pancreatic cancer	1–2	60	China
**NCT02041936**	Evaluation of the short and intermediate term outcomes of ablation of locally advanced unresectable pancreatic cancer using the Nanoknife IRE System—A prospective study	NA	12	USA
**NCT03257150**	IRE for locally advanced pancreatic ductal adenocarcinoma (LEAP TRIAL): A phase I/II prospective trial	1–2	47	Canada
**NCT03105921**	IRE (Nanoknife) for the Treatment of Pancreatic Adenocarcinoma	NA	20	France
**NCT02674100**	AHPBA Pancreatic Irreversible electroporation (IRE) registry for pancreatic cancer	NA	500	USA, Japan, Taiwan, UK

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
