# Peer review of "Systematic Review of Irreversible Electroporation Role in Management of Locally Advanced Pancreatic Cancer"

_cancers, 2019, doi:10.3390/cancers11111718_

Round 1
Reviewer 1 Report
Dear Authors, I read with interest your paper which is timely since growing attention to ablative treatments for pdac.
Revision of the literature is satisfactory and the topic is well discussed
Author Response
Thank you for your comments.
Reviewer 2 Report
The authors review the literature for the use of IRE in locally advanced pancreatic cancer and find 15 studies with eligible to be included. The authors summarize the studies in table 1 and include mOS and mortality rate. This is an important topic to perform a literature review on, however, the review is poorly written and contains numerous grammatically errors. Additionally, I would recommend the following:
1. Indicate "review" in the title of the manuscript
2. Additional information on patient cases (median age, prior chemotherapy, prior radiation, etc.) would be important to include if available.
3. Include a list/table of current clinical trials utilizing IRE in pancreas cancer
Author Response
Thank you for your comments and suggestions.
Point 1. Indicate "review" in the title of the manuscript
Response 1: I have changed the title in "Systematic review of Irreversible electroporation role in management of Locally Advanced Pancreatic Cancer"
Point 2. Additional information on patient cases (median age, prior chemotherapy, prior radiation, etc.) would be important to include if available.
Response 2: I have extended the information on patient cases in lines 161-162, 167-172 and also in Table 2
Point 3. Include a list/table of current clinical trials utilizing IRE in pancreas cancer
Response 3: I have created a table of current clinical trials utilizing IRE in pancreas cancer (table 3)
Reviewer 3 Report
Comments to Authors:
The review of literature on Irreversible Electroporation (IRE) in locally advanced pancreatic cancer (LAPC) was extensive and thorough. I have a few recommendations to make the article more reader friendly.
Lines 38-39 talks about percentage of PC patients who are candidates for surgical resection. The number is very low. Please comment on why this is and move the description on what makes a candidate suitable for resection (from Lines 48-64) together with these numbers Line 71: what is the current survival rate with Gemcitabine. Compare this number with 24 months with FOLFIRINOX. What is IRE and the explanation of the procedure should be moved to the introduction. This is important to set the stage for the study.Currently this is written in the discussion. Include another table with pros and cons of IRE vs other ablative methods. How might we alleviate the inflammatory responses/pancreatitis that may be caused by IRE? Include a paragraph in the discussion detailing on this topicMinor comment: English needs to be improved and grammatical mistakes corrected in the article
Author Response
Thanks for your review,
Point 1. Lines 38-39 talks about percentage of PC patients who are candidates for surgical resection. The number is very low. Please comment on why this is
Response 1: I have fixed it and reorganized the text in lines 40-42 (in red)
Point 2. Move the description on what makes a candidate suitable for resection (from Lines 48-64) together with these numbers
Response 2: after describing LAPC in lines 48-64, I added in the lines 66-67 (in red) the concept ' what makes a candidate suitable for resection'. I don't think it is possible to be more specific, because as described, the concept of 'resecability' is not yet unanimous.
Point 3. Line 71: what is the current survival rate with Gemcitabine. Compare this number with 24 months with FOLFIRINOX.
Response 3: Fixed in lines 73-75
Point 4. What is IRE and the explanation of the procedure should be moved to the introduction. This is important to set the stage for the study.Currently this is written in the discussion.
Response 4: Fixed in lines 104-124 (in red).
Point 5. Include another table with pros and cons of IRE vs other ablative methods.
Response 5: Table 1
Point 6. How might we alleviate the inflammatory responses/pancreatitis that may be caused by IRE? Include a paragraph in the discussion detailing on this topic
Response 6: Fixed in lines 249-255
Point 7. English needs to be improved and grammatical mistakes corrected in the article
Response 7: English corrected